# Optimizing cloud optical parameterizations in Radiative Transfer for TOVS (RTTOV v12.3) for data assimilation of satellite visible reflectance data: an assessment using observed and synthetic images

Yongbo Zhou<sup>1</sup>, Tianrui Cao<sup>1</sup>, Lijian Zhu<sup>2</sup>

<sup>1</sup>School of Atmospheric Physics, Nanjing University of Information Science & Technology, Nanjing, China <sup>2</sup>Shanghai Typhoon Institute, China Meteorological Administration, Shanghai, China

Correspondence to: Yongbo Zhou (yongbo.zhou@nuist.edu.cn)

Abstract. The Radiative Transfer for TOVS (RTTOV) is a commonly used forward operator software package for the Data Assimilation (DA) of satellite visible reflectance data. However, a wide choice of Cloud Optical Parameterizations (COPs) in RTTOV poses challenges in discerning the optimal configuration. In this study, the performance of different COPs was evaluated by comparing the observed and synthetic visible satellite images. Observed images (O) were provided by Fengyun (FY)-4B and Himawari-9, two operational geostationary meteorological satellites covering East Asia. Synthetic images (B) were generated by RTTOV (v12.3) with the Discrete Ordinate Method (DOM) and the Method for FAst Satellite Image Simulation (MFASIS). The inputs to RTTOV were provided by the 3-h forecasts of the China Meteorological Administration Mesoscale (CMA-MESO) model and the fifth generation European Centre for Medium-Range Weather Forecasts reanalysis (ERA5) data. On the domain average, B was smaller than O, especially in cloudy situations. The minimum O-B bias was revealed for the COP of liquid water clouds in terms of effective diameter (Deff) in combination with the COP of ice clouds developed by the Space Science and Engineering Center (SSEC), with the Deff for ice clouds parameterized in terms of ice water content and temperature. Compared with the O-B biases, the standard deviations of the O-B departure were less sensitive to COPs. In addition, histogram analysis of reflectance indicated that the synthetic images with the minimum O-B bias resembled best with the observed images. Therefore, the optimal cloud optical parameterization was proposed to be the "Deff" + "SSEC" suite.

## 1 Introduction

The Fengyun (FY) -4A and -4B are two China's new-generation geostationary meteorological satellites that form a dual-satellite observation constellation. The two satellites carry an Advanced Geostationary Radiation Imager (AGRI), which provides radiance observations ranging from visible to infrared bands over East Asia and Western Pacific areas. From February 1 to 5 March 2024, FY-4B replaced the FY-4A and started its operational observations from 00:00 UTC on 5 March 2024. The spectral observations contain vital information on cloud, precipitation, aerosols, underlying surfaces, temperature, and

humidity (Yang et al., 2017). Compared with infrared radiation, visible radiation is capable of penetrating deeper into the cloud and thereby provides more information therein. In addition, visible radiation is less sensitive to humidity and temperature, which facilitates the remote sensing of lower-layer clouds. Therefore, visible radiance data is receiving attention from DA community (Scheck et al., 2020; Schröttle et al., 2020; Zhou et al., 2022, 2023; Kugler et al., 2023).

For the DA of satellite radiance data, it is necessary to convert the model state into simulated observations to compute innovations. The accuracy of the forward operators influences the analysis fields and the subsequent forecasting results in two aspects. On one hand, forward operators influence the observation increments, which are converted into the analysis increments in the state variable space (e.g., Anderson, 2001). On the other hand, forward operators influence the bias correction of satellite radiance data because the bias correction is usually based on observation (O) minus background (B) analyses, where B is simulated by a forward operator based on the forecasts of a numerical weather prediction (NWP) model (Harnisch et al., 2016; Noh et al., 2023; Zhou et al., 2024). DA of FY-4A visible radiance data has been explored under the framework of Observing System Simulation Experiment (OSSE) (Zhou et al., 2022, 2023). For the experiment designs of OSSE, the model equivalents of observation were simulated by the Radiative Transfer for TOVS (RTTOV) using the Discrete Ordinate Method (DOM) (RTTOV-DOM hereafter). Meanwhile, the background model state was converted into the simulated observations by the same forward operator. Therefore, errors due to the forward operator were neglected. However, to extend the DA of FY-4A and 4B visible reflectance to real-world cases, the performance of the forward operator must be thoroughly evaluated.

Recently, the O-B statistics of FY-4A visible channel (0.55 - 0.75  $\mu$ m) were explored using RTTOV-DOM based on the short-term forecasts of the China Meteorological Administration Mesoscale (CMA-MESO) (Zhou et al. 2024). One of the findings of Zhou et al. (2024) is that the synthetic images were more reliable in cloud-free situations than for the cloudy situations. In cloudy situations, an important factor constraining the performance of the RTTOV is the cloud optical parameterizations. In Zhou et al. (2024), cloud optical properties were estimated by the ice cloud optical parameterization developed by Baran et al. (2014) (the "Baran 2014" parameterization) and the liquid water cloud optical parameterization in terms of effective diameter ( $D_{eff}$ ) (the " $D_{eff}$ " parameterization) (Hocking et al., 2019). Considering that a wide choice of liquid water cloud and ice cloud optical parameterizations was available in RTTOV, the configuration of cloud optical parameterizations in Zhou et al. (2024) exhibited a certain degree of blindness.




The cloud optical parameterizations in RTTOV were developed based on theoretical computations and in-situ measurements of cloud microphysical properties such as the Particle Size Distribution (PSD), the shapes and mixing ratios of different ice habits (for ice clouds only), etc (Baum et al., 2011). It is possible that there are discrepancies in cloud microphysical properties between the built-in parameterizations and NWP models (or real cases). As a result, the performance

of RTTOV in synthetizing visible satellite images should vary with the configuration of different cloud optical parameterizations.

In order to provide guidance for optimizing the configuration of cloud optical parameterizations in RTTOV for the DA of the satellite visible reflectance data, the performance of RTTOV configured with all available liquid water cloud and ice cloud optical parameterizations was evaluated by comparing the observed and the synthetic visible satellite images. The remaining part of this manuscript is organized as follows. Cloud optical parameterizations are introduced in Section 2. Experiment designs, data, and method are presented in Section 3. Results for the reference experiment are shown and discussed in Section 4. Discussions on the generalization of the results for different experiment designs and comparison with a previous study are presented in Section 5. Conclusions are summarized in Section 6.

## 2 Cloud optical parameterizations in RTTOV


All available cloud optical parameterizations built-in the RTTOV (version 12.3) (Saunders et al., 2018) were summarized in Table 1. For the liquid water cloud and ice cloud, cloud optical parameterizations were further divided into several subtypes, which were illustrated in Section 2.2.1 and Section 2.2.2, respectively.

Table 1. The liquid water cloud parameterization (clw\_scheme) and ice cloud parameterization (ice\_scheme) in the version 12.3 of RTTOV software package (Hocking et al., 2019).

| Cloud types        | Name of cloud optical parameterization | Settings in RTTOV |
|--------------------|----------------------------------------|-------------------|
| liquid water cloud | OPAC                                   | clw_scheme = 1    |
|                    | $D_{eff}$                              | clw_scheme = 2    |
| ice cloud          | SSEC                                   | ice_scheme = 1    |
|                    | Baran 2014                             | ice_scheme = 2    |
|                    | Baran 2018                             | ice_scheme = 3    |

#### 2.1 Liquid water cloud optical parameterizations

#### 2.1.1 The OPAC parameterization

The Optical Properties of Aerosols and Clouds (OPAC) parameterization in RTTOV provides optical properties for five liquid water cloud types, including the continental stratus (STCO), the maritime cumulus (STMA), the clean continental stratus

(CUCC), the polluted continental stratus (CUCP), and the maritime cumulus (CUMA) (Hess et al., 1998). For each of the five cloud types, the PSD was described by a pre-defined modified Gamma function, and the  $D_{eff}$  determined by the pre-defined PSD was 8.0  $\mu$ m, 14.6  $\mu$ m, 11.5  $\mu$ m, 22.6  $\mu$ m, and 25.3  $\mu$ m for CUCP, STCO, CUCC, STMA, and CUMA, respectively. The volumetric optical properties for unit cloud concentration (1 g m<sup>-3</sup>) were calculated by integrating the optical properties derived from the Mie theory over the PSDs. In the RTTOV simulations, the extinction and scattering coefficients of liquid water cloud are computed as the product of the respective per-unit-concentration coefficients and the total liquid hydrometeor concentration (comprising cloud droplets and raindrops for the CMA-MESO forecasts).

To generate synthetic visible satellite images, the atmospheric fields were processed into the format of the RTTOV input files for each of the five liquid cloud types following the flowchart shown by Figure 1. A pre-trail experiment indicates that the differences of simulated reflectance between CUCC and CUCP were at the order of 10<sup>-3</sup>. Therefore, liquid water cloud over land was set to CUCC.

Figure 1. The flowchart for assigning one of the five OPAC liquid water cloud subtypes from the atmospheric fields.

# 2.1.2 The " $D_{\it eff}$ " parameterization


Cloud optical properties for the " $D_{\it eff}$ " parameterization were parameterized in terms of  $D_{\it eff}$  and the mixing ratios of liquid water cloud hydrometeors. Since the atmospheric fields, as will be introduced in Section 3.2.1 and Section 3.2.2, did not include  $D_{\it eff}$ ,  $D_{\it eff}$  was estimated by Equation (1) explicitly (Thompson et al., 2004),

$$D_{eff} = \left(\frac{1.91\rho_a q_w}{\rho_w N_w}\right)^{1/3} \tag{1}$$

where  $\rho_a$  and  $\rho_w$  denote the density of air and cloud water droplets, respectively.  $q_w$  denotes the mixing ratio of cloud droplets, and  $N_w$  denotes the number concentration of cloud droplets.  $\rho_a$  and  $q_w$  were derived from the CMA-MESO forecasts. In this study,  $\rho_w$  was set to 1000  $kgm^{-3}$ . In addition,  $N_w$  was set to 300  $cm^{-3}$ , which is the default number concentration of cloud condensation nuclei (CCN) in the CMA-MESO model configured with the single-moment six-class microphysical scheme (the default physic scheme for operational forecasting). The presumed CCN number concentration may influence the overall results, and this potential impact was discussed in Section 5.5.

Following the parameterization scheme of Thompson et al. (2004), raindrops were formed from cloud droplets through an autoconversion process involving collision and coalescence. Therefore, Equation (1) did not explicitly account for the impact of raindrops on  $D_{\it eff}$ . Nevertheless, the mixing ratios of cloud droplets and raindrops were summed (representing total liquid hydrometeor concentration) to capture the influence of liquid cloud hydrometeors on the radiative transfer simulations.

## 105 **2.2 Ice cloud optical parameterizations**

## 2.2.1 The "SSEC" parameterization

For the ice cloud optical parameterization developed by the Space Science and Engineering Center (SSEC) (the "SSEC" parameterization), the ice cloud optical properties were parameterized in terms of IWC and  $D_{\it eff}$ .  $D_{\it eff}$  was not provided by the RTTOV users explicitly. Instead,  $D_{\it eff}$  was estimated by the following four built-in parameterizations.

By the parameterization of Ou and Liou (1995) (OL95 hereafter),  $D_{eff}$  was parameterized in terms of ambient temperature  $T_c$  in Celsius,

$$D_{eff} = 326.3 + 12.42T_c + 0.197T_c^2 + 0.0012T_c^3$$
 (2)

By the parameterization of Wyser (1998) (W98 hereafter),  $D_{eff}$  was parameterized in terms of ambient temperature T in Kelvin and IWC (unit:  $gm^{-3}$ ),

$$D_{eff} = 377.4 + 203.3B + 37.91B^2 + 2.3696B^3$$
 (3)

where B is described by Equation (4),

$$B = -2 + 10^{-3} (271 - T)^{1.5} \log_{10} (\text{IWC} / 50)$$
(4)

By the parameterization of Boudala et al. (2002) (B02 hereafter),  $D_{eff}$  was parameterized in terms of  $T_c$  and IWC,

$$D_{eff} = 53.005 IWC^{0.06} \exp(0.0013T_c)$$
 (5)

By the parameterization of McFarquar et al (2003) (MF03 hereafter),  $D_{eff}$  was parameterized in terms of *IWC*,

$$D_{eff} = 2 \times 10^{1.78449 + 0.281301 \log(IWC) + 0.01777166 [\log(IWC)]^2}$$
(6)

Apart from the four built-in parameterizations of  $D_{e\!f\!f}$  for ice cloud, we evaluated an extra parameterization in terms of the number concentration of ice nuclei (IN) (described by Equations (2) – (5) in Yao et al., 2018). However, the IN-related parameterization did not generate a better performance than the four built-in parameterizations. Therefore, the results for the IN-related parameterization were not shown for simplicity.

Similar to the liquid water clouds, the extinction and scattering coefficients of ice cloud are computed as the product of the respective per-unit-concentration coefficients and the total ice hydrometeor concentration. The total ice hydrometeor concentration is the sum of the concentration for ice, snow, and graupel for the CMA-MESO forecasts.

## 2.2.2 The "Baran" parameterization


For the "Baran" parameterization, the ice cloud optical properties were parameterized in terms of IWC and temperature (Vidot et al., 2015). Unlike the "SSEC" parameterization, the "Baran" parameterization does not have dependence on  $D_{eff}$ . The old "Baran" parameterization (Baran 2014) was updated (Baran 2018) to smooth the discontinuous variation of absorption and scattering coefficients within the shortwave spectral range (Saunders et al., 2020). Therefore, Baran 2018 is more spectrally consistent than Baran 2014 and as such it is recommended over the latter by Hocking et al. (2019).

Table 2. Different combination of cloud optical parameterizations. clw\_scheme and ice\_scheme denote the liquid water cloud and ice cloud optical parameterizations, respectively. "idg" denotes the built-in parameterization of  $D_{eff}$  for ice cloud. idg=1: OL95, idg=2: W98, idg=3: B02, idg=4: MF03.

| clw_scheme | ice_scheme | idg | Name for combination of different parameterizations |
|------------|------------|-----|-----------------------------------------------------|
| 1          | 1          | 1   | C111                                                |
| 1          | 1          | 2   | C112                                                |
| 1          | 1          | 3   | C113                                                |
| 1          | 1          | 4   | C114                                                |
| 1          | 2          |     | C12n                                                |
| 1          | 3          |     | C13n                                                |
| 2          | 1          | 1   | C211                                                |
| 2          | 1          | 2   | C212                                                |
| 2          | 1          | 3   | C213                                                |
| 2          | 1          | 4   | C214                                                |
| 2          | 2          |     | C22n                                                |
| 2          | 3          |     | C23n                                                |

In summary, there are 12 combinations of cloud optical parameterizations (including the four built-in parameterizations of  $D_{eff}$  for ice cloud) for liquid water cloud and ice cloud (Table 2). The performance of different combinations of cloud optical parameterizations was evaluated by the experiment designs introduced in Section 3.1. Based on the configuration of cloud optical parameterizations, radiative transfer simulations were performed by the DOM solver and the Method for FAst Satellite Image Simulation (MFASIS). For the DOM solver, 16 streams were used. The general radiative transfer options account for atmospheric refraction and curvature. The surface Bidirectional Reflectance Distribution Function (BRDF) was either derived from monthly mean atlases for land surface (Vidot and Borb &, 2014; Vidot et al., 2018) or calculated by the JONSWAP (Hasselmann et al., 1973) solar BRDF model for sea surface. Since aerosol variables were not provided by the atmospheric fields, the contribution of aerosols to visible reflectance was neglected during the radiative transfer simulations. Nevertheless, the aerosol impact on the evaluation results was discussed in Section 5.6.

#### 3. Experiment designs, data, and method

## 3.1 Experiment designs




The performance of different cloud optical parameterizations was evaluated by four experiments summarized in Table 3. The experiments cover different observing systems (the visible bands of FY-4B and Himawari-9, where the latter is the operational geostationary meteorological satellite operated by the Japan Meteorological Agency), different atmospheric fields (the CMA-MESO forecasts and the fifth generation European Centre for Medium-Range Weather Forecasts reanalysis (ERA5) data), and different radiative solvers (DOM and MFASIS) to generalize the main findings.

Table 3. The four experiments covering different atmospheric fields, observing systems, and radiative solvers.

| Atmospheric fields | Observing systems | solver in RTTOV | Experiment name |
|--------------------|-------------------|-----------------|-----------------|
| CMA-MESO forecasts | FY-4B band 2      | DOM             | CM-FY-DM        |
| ERA5 data          | FY-4B band 2      | DOM             | E5-FY-DM        |
| CMA-MESO forecasts | Himawari-9 band 3 | DOM             | CM-HW-DM        |
| CMA-MESO forecasts | Himawari-9 band 3 | MFASIS          | CM-HW-MF        |

The comparison was performed for the band 2 (centering at  $0.65 \mu m$ ) of AGRI onboard FY-4B and the band 3 (centering at  $0.64 \mu m$ ) of the Advanced Himawari Imager (AHI) onboard Himawari-9. The Himawari-9 band 3 was chosen due to the spectral and spatiotemporal matches with the FY-4B visible band. On one hand, the visible bands of the two satellites share

considerable similarities with respect to spectral characteristics (Figure 2). Therefore, results derived from the two instruments should have certain consistency. On the other hand, FY-4B and Himawari-9 are above the equator at the longitudes of 104.7  $\times$  and 140.7  $\times$ , respectively. In addition, the full-disk scanning cycles of AGRI and AHI are 15 min and 10 min, respectively. Therefore, the observation areas and times of the two satellites exhibit substantial overlaps.

Figure 2. The spectral response functions for the band 2 of AGRI onboard FY-4B and for the band 3 of AHI onboard Himawari-9.

MFASIS is a Look-up Table (LUT)-based emulator of a 1D solver (Scheck et al., 2018). When constructing the LUT for the MFASIS, radiative transfer simulations were also affected by the cloud optical parameterizations. Therefore, synthetic images generated by the DOM and MFASIS solvers should reveal some common characteristics in terms of the cloud optical parameterizations built-in the RTTOV software package.

Each of the four experiments includes 12 configurations of cloud optical parameterizations (Table 2), except for the CM-HW-MF experiment which only involved the "SSEC" ice cloud optical parameterization. In fact, MFASIS was initially designed for the Spinning Enhanced Visible and Infrared Imager (SEVIRI) onboard the European Organisation for the Exploitation of Meteorological Satellites (EUMETSAT). For the version 12.3 of RTTOV, MFASIS was extended to the Himawari-9 observing systems, but only the "SSEC" ice cloud optical parameterization was available.

#### 3.2 Data





#### 3.2.1 The 3-h forecasts of CMA-MESO model

The CMA-MESO forecasts from March 15th to April 25th, 2024 were used to provide the inputs to RTTOV. The CMA-MESO model is a cycled DA and forecasting system. The model was initialized eight times per day at 00:00, 03:00, 06:00, 09:00,

12:00, 15:00, 18:00, and 21:00 UTC. At 00:00 and 12:00 UTC, the model was cold-started, with the Initial Conditions (ICs) and Lateral Boundary Conditions (LBCs) provided by the CMA-Global Forecasting System (GFS). At other startup times, the model was warm-started, with the ICs and LBCs updated from the analysis fields that were generated by a cloud analysis technique and by assimilating synergic observations with a 3D variational (3DVar) DA method (Shen et al., 2020).

To avoid low solar elevation, only the 3-h forecasts at 06:00 UTC were selected. The 3-h forecasts were chosen based on an evaluation study of CMA-MESO model (originally termed the GRAPES\_3km model) by Zhang et al. (2020). The study suggested that the CMA-MESO forecasts were much more reliable within the initial three forecasting hours, and the "spin-up" issue was essentially absent with the incorporation of a cloud analysis technique. The domain coverage of the CMA-MESO model includes 2501 × 1671 horizontal grids with a grid spacing of 0.03° and 50 vertical layers with a model top of 10 hPa. To avoid 3D radiative effects on high-resolution radiative transfer simulations, the CMA-MESO forecasts were superobbed to a horizontal resolution of 0.09° with 833×557 horizontal grids. The superobbing was performed by simply averaging the 0.03°×0.03° products every three grids. In the following, the CMA-MESO forecasts would refer to the 0.09°×0.09° products unless otherwise specified.

#### 3.2.2 The ERA5 data





The 0.25 °×0.25 ° gridded ERA5 data (Hersbach et al., 2020) on pressure levels (Hersbach et al., 2018a) and on single levels (Hersbach et al., 2018b) were used to provide the inputs to RTTOV. The ERA5 data on pressure levels were generated by the ECMWF Integrated Forecast System (IFS) based on worldwide observations using a 4DVar DA method. The pressure-level data provide temperature, water vapor mixing ratio, and cloud-related parameters (the mixing ratio of cloud, ice, rain, and snow, as well as cloud cover) on 137 pressure levels. In addition, the ERA5 data on surface level were generated by a coupled model of the atmospheric model in the IFS and a land-surface model. The surface-level data provide the 2-m (and 10-m) wind, 2-m (and 10-m) temperature, 2-m (and 10-m) humidity, and surface pressure, etc.

## 3.2.3 FY-4B data

The FY-4B 4 km×4 km visible reflectance data, cloud mask (CLM) product, and the synchronous observation geometry (GEO) data were used. To generate spatially collocated observed and synthetic images, the FY-4B full-disk visible reflectance data were horizontally averaged to the locations of the CMA-MESO forecasts (or ERA5) grids. The horizontal averaging was performed by the following two steps. First, centering at a given CMA-MESO (or ERA5) grid and finding all the pixels (matched pixels hereafter) in the FY-4B/AGRI visible image within ±0.045 ° for the 0.09 °×0.09 ° CMA-MESO forecasts (or

within ±0.125 °for the 0.25 °×0.25 °ERA5 data) both in the zonal and meridional directions. Second, averaging the reflectances of all these matched pixels to generate a reflectance that is spatially matched to the selected CMA-MESO (or ERA5) grid. Repeating the two steps for all CMA-MESO (or ERA5) grid points generated an observed image gridded at 0.09 °×0.09 ° (or 0.25 °×0.25 °). The full-disk scanning cycle of FY-4B/AGRI is 15 minute and the scanning starts at 00:00 UTC. In addition, the CMA-MESO forecasts and ERA5 data were produced at hourly intervals. Therefore, the maximum allowable time difference between the FY-4B observations and CMA-MESO forecasts and ERA5 data was within 15 minute to ensure a temporal match.

To facilitate the radiative transfer simulations by RTTOV, the sun-viewing geometries (i.e., solar zenith and azimuth angles, satellite zenith and azimuth angles) were derived from FY-4B GEO data. The 4 km ×4 km GEO data were horizontally averaged to the CMA-MESO (or ERA5) grids in the same way described above. In addition, the FY-4B CLM product was used to provide a first-step estimate of cloud mask for an arbitrary CMA-MESO or ERA5 grid. The 4 km ×4 km CLM product was matched to the CMA-MESO (or ERA5) grids by a maximum occurrence method, i.e., cloud mask for a CMA-MESO (or ERA5) grid was set to that with the maximum occurrence frequency amongst the entire matched FY-4B pixels.

#### 3.2.4 Himawari-9 data


The Himawari-9 5 km×5 km products, including the reflectance at band 3, the GEO data, and cloud type product, were matched to the CMA-MESO and ERA5 grids in the same way introduced by Section 3.2.3. The Himawari-9 cloud type product not only provided information on cloud or clear sky for a certain pixel, but also on the cloud types for cloudy scenarios. Clouds were divided into eight subtypes, including cirrus (Ci), cirrostratus (Cs), deep convection (Dc), altocumulus (Ac), altostratus (As), nimbostratus (Ns), cumulus (Cu), and stratus (Sc).

## 3.3 Equivalent criteria of cloud mask for the observed and synthetic images

Since statistical characteristics of the O-B departure are different for cloudy and clear pixels, it is critical to evaluate the results for the two scenarios separately. To ensure equivalent criteria of cloud mask for the observed and synthetic images, the observed and synthetic visible images were compared with the images simulated by ignoring cloud impacts (Zhou et al., 2024). For synthetic images, a pixel was designated to be cloudy if Equation (7) was satisfied. Otherwise, the pixel was designated to be cloud-free.

$$230 r_{sim} > r_{clr} (7)$$

where  $r_{sim}$  denotes the reflectance for an arbitrary pixel in a synthetic image, and  $r_{clr}$  denotes the spatiotemporally collocated reflectance simulated by ignoring cloud impacts.

The aerosol contributions were neglected by the RTTOV simulations. However, the observed reflectance inevitably included aerosol contributions. To account for the aerosol impacts on the cloud masking for the observed images, a pixel was designated to be cloudy if the observed reflectance  $r_{obs}$  satisfied Equation (8),

$$r_{obs} > r_{sim} + r_{aer}^{75} \tag{8}$$

where  $r_{aer}^{75}$  denotes the aerosol contribution to the reflectance for cloudy pixels.  $r_{aer}^{75}$  was set to the upper quartile of  $r_{obs} - r_{clr}$  for the preliminarily estimated cloud-free pixels, which were designated by the cloud mask derived from the FY-4B CLM product. The second-step estimate of cloud-free pixels was determined Equation (9),

$$240 r_{obs} 

Figure 3. Time series of the O-B biases and standard deviations for (a-b) all, (c-d) cloudy, and (e-f) clear pixels. The results are for the CM-FY-DM experiment.

The maximum difference of the standard deviations for different cloud optical parameterizations was within 0.03. The probability density distribution function (PDF) of reflectance for the synthetic images showed over- or underestimation (compared with the observed images) in the occurrence frequency at low (<0.03) and moderate-to-large (>0.2) reflectance (Figure 4). In comparison, the PDF for C112, which was taken as an example for the sub-optimal cloud optical parameterization, was also shown by Figure 4. The PDF for C213 resembles that for the observation better than other cloud optical parameterizations (not shown for simplicity). Since the PDFs were not sensitive to the location of the clouds (Geiss et al., 2021), the PDF analysis suggested an overall improvement compared with other cloud optical parameterizations.




Figure 4. The Probability density Distribution Functions (PDFs) of reflectance for the observed and synthetic images from March 15 to April 15 in 2024.

To illustrate the reasons to the performance of different cloud optical parameterizations, the sensitivity study of reflectance to different cloud optical parameterizations was demonstrated by Figure 5. According to the Mie theory, the larger the cloud particle size, the stronger the forward scattering, and vice versa. Therefore, the largest (smallest) reflectance was expected for the CUCP (CUMA) cloud subtype. The " $D_{eff}$ " parameterization generated larger TOA reflectance than the "OPAC" scheme, which implies that the backscattering effects of the " $D_{eff}$ " parameterization are stronger than the "OPAC" parameterization.

Figure 5. Dependence of FY-4B/AGRI band 2 reflectance on cloud water path (CWP) for different (a) liquid water cloud optical parameterizations and (b) ice cloud optical parameterizations. For this simulation, the solar zenith angle, viewing zenith angle, and relative azimuth angle are set to  $25^{\circ}$ ,  $40^{\circ}$ , and  $135^{\circ}$ , respectively.

For the ice cloud optical parameterizations, Baran 2014 outperformed Baran 2018 when measured by the O-B biases. The mean volumetric scattering coefficients of Baran 2014 were larger than the Baran 2018 (Saunders et al., 2020). Therefore, more photons were backscattered for the Baran 2014 parameterization. The results in Figures 3 and 5 suggested that the Baran 2018 ice scheme should be used with caution for the radiative transfer simulations in visible spectral ranges. In addition, the performance of the "SSEC" ice cloud optical parameterization was sensitive to the parameterization of  $D_{eff}$  for ice clouds. An inter-comparison between the four parameterizations of  $D_{eff}$  was illustrated by Figure 6, which revealed that the minimum (maximum) effective diameter was found for the B02 (W98) parameterization. Therefore, the largest (smallest) reflectance was expected for the B02 (W98) parameterization.




The optical characteristics of the "Baran 2014 + B02" ice cloud parameterization may partially explain the observed phenomenon where the C213 parameterization successfully corrects the systematic underestimation of B while introducing greater standard deviation (or local variability) (Figure 3a-d). Within the same cloud water path (CWP) range, C213 produces the largest variation amplitude in B. Notably, when transitioning from cloud-free to cloudy conditions, the "Baran 2014 + B02" scheme exhibits the most significant reflectance variation range (Figure 5b). This enhanced variability in B could potentially broaden the distribution of O-B discrepancies, thereby increasing the standard deviation of O-B. However, it's important to note that the standard deviations of the O-B departure showed less sensitivity to cloud optical parameterizations compared to the biases.

Figure 6. (a) The effective diameter estimated by the four built-in parameterizations in RTTOV. (b) The vertical distribution of temperature and ice water content for the sensitivity study shown by Figure 5.

## 4.2 Spatial distribution of the O-B departure



One-month O-B departure statistics over the study domain reveal systematic errors and aid in result understandings. The spatial distribution of the O-B biases, the standard deviations of the O-B departure, and the correlation coefficients between O and B were derived from the one-month observed and synthetic images (Figure 7). Compared with O, B was underestimated over the Siberia, the Mongolian Plateau, the Southern foothills of the Himalayas, the Sichuan basin, and the Yunnan-Kweichow Plateau (Figure 7(a)). In general, the spatial distribution of the correlation coefficients between O and B agreed well with the spatial distribution of the O-B biases (Figure 7(c)). Namely, small O-B biases agreed well with large correlation coefficients. The Siberia, the Mongolian Plateau, the Southern foothills of the Himalayas were covered with snow in March and April. Reflectance simulated in these areas should be less accurate compared with other places because the BRDF atlas is questionable in snow-covered areas (Ji et al., 2022). Since the areas with O-B bias larger than zero were in good agreement with the areas potentially being covered by snow (Figure 8), a tentative conclusion could be drawn that the surface albedo (equivalent to the BRDF for Lambertian radiators) over the snow-covered areas was underestimated.

Figure 7. (a) Spatial distribution of the one-month biase of O-B departure for the CM-FY-DM experiment; (b) Spatial distribution of the one-month standard deviation of O-B departure; (c) Spatial distribution of the one-month correlation coefficient between O and B.

Figure 8. (a) Spatial distribution of the O-B biases for collocated cloud-free pixels from March 15 to April 15 in 2024; (b) Spatial distribution of the mean BRDF from March 15 to April 15 in 2024.

Figure 9. (a) Synthetic FY-4B/AGRI channel 2 visible image generated by RTTOV for the C213 configuration at 06:00 UTC on 25 March 2024. (b) Synthetic FY-4B/AGRI channel 2 visible image generated by RTTOV for the C13n configuration. (c) Spatiotemporally collocated visible image observed by FY-4B/AGRI channel 2.

In addition, the performance of the CMA-MESO model was reduced in some circumstances. An example for the observed and synthetic images at 06:00 UTC, March 25th 2024 was shown by Figure 9. The selected case reported a typical commashaped cloud which was caused by a frontal system. In general, B was undervalued in some parts of Eastern China and the Western Pacific areas (Figure 9(c)) which were classified to be convective clouds (Figure 9(d)). Similar results were reported by Zhou et al. (2024). A potential explanation is the deficiency of the CMA-MESO model in forecasting the strongly convective weather systems (Wan et al., 2015). In addition, some of the Ci and Cs clouds were missed or underestimated over the Himalayas, the Sichuan basin, and the Yunnan-Kweichow Plateau. This is most likely caused by the reduced performance of the CMA-MESO model over complex terrain areas (Wan et al., 2015; Zhu et al., 2017).

#### 5. Discussions





## 5.1 Results for E5-FY-DM: influences of atmospheric fields

The comparison between CM-FY-DM and E5-FY-DM experiments was designed to reveal the influences of different atmospheric fields on the statistical characteristics of O-B departure. The results are shown by Figure 10.

In general, the results for E5-FY-DM are similar to these for CM-FY-DM. To be specific, the minimum O-B bias was revealed for the cloud optical parameterization of C213, and the standard deviations were less sensitive to cloud optical parameterizations compared with the O-B biases. However, the biases and standard deviations of O-B departure were smaller than those for the CM-FY-DM experiment (Figure 3). There are two potential explanations. On one hand, the coarser grids for the E5-FY-DM experiment meant more averaging of the atmosphere fields and the subsequent reflectance fields. The horizontal averaging tended to smooth the reflectance fields, i.e., the larger reflectance tended to be reduced, and vice versa. As a result, the PDF of the O-B departure shrank, and the differences between O and B were reduced. On the other hand, it was possible that atmospheric fields, especially cloud variables, were better represented by the ERA5 data than the CMA-MESO forecasts. Since errors in B were mainly determined by atmospheric fields and the forward operators, consistent results for different atmospheric fields increased the robustness of the findings in Section 4.

Figure 10. Time series of the O-B biases and standard deviations for (a-b) all, (c-d) cloudy, and (e-f) clear pixels. The results are for the E5-FY-DM experiment.

# 5.2 Results for CM-HW-DM: influences of observing systems



The comparison between CM-FY-DM and CM-HW-DM experiments was designed to reveal the influences of different observing systems on the statistical characteristics of the O-B departure. The results for the CM-HW-DM experiment are shown by Figure 11.

In general, the results for the CM-HW-DM experiment were similar to these for the CM-FY-DM. However, the O-B biases and standard deviations were smaller for the CM-HW-DM experiment than the CM-FY-DM. Although the centering wavelengths of the FY-4B and Himawari-9 visible bands are close, the spectral response function has a wider range for the FY-4B visible band than the Himawari-9 visible band (Figure 2). As a result, the convolution of monochromatic cloud optical properties over the spectral response function would involve a broader wavelength range. If cloud optical parameterizations

contain errors, it is likely that including broader wavelength range would amplify the errors in volumetric optical properties, leading to larger O-B biases and standard deviations. In addition, the differences of the O-B departure between the two observing systems could be related to the measurement calibration processes. The radiometric calibration techniques for the two visible bands were performed by different methods (Okuyama et al., 2018; Zhang et al., 2024). Since 29 May 2023, the National Satellite Meteorological Center (NSMC) has not updated the calibration coefficients of the FY-4B/AGRI solar reflection bands. It is possible that the radiometric performance of the instrument was declined due to the influences of space particle erosion, device aging, etc.



Figure 11. Time series of the O-B biases and standard deviations for (a-b) all, (c-d) cloudy, and (e-f) clear pixels. The results are for the CM-HW-DM experiment.

#### 5.3 Results for CM-HW-MF: influences of radiative solvers



The comparison between CM-HW-MF and CM-HW-DM experiments was designed to reveal the influences of different radiative solvers on the statistical characteristics of O-B departure. The results for the CM-HW-MF experiment are shown by Figure 12.

Figure 12. Time series of the O-B biases and standard deviations for (a-b) all, (c-d) cloudy, and (e-f) clear pixels. The results are for the CM-HW-MF experiment.

In general, similar results were revealed between the two experiments. However, the O-B biases and standard deviations were smaller for the CM-HW-MF (Figure 12) than the CM-HW-DM. Since the B was smaller than O on the domain average, the reflectance simulated by the MFASIS was slightly larger than that simulated by the DOM solver. This is most likely related to the differences between the two solvers in tackling the scattering interactions between clouds and gaseous molecules. The multiple scattering processes in MFASIS were considered for all cloudy and clear layers (Scheck et al., 2016). However for DOM solver, the multiple scattering processes were only limited to cloudy layers, and the scattering interactions between clouds and gaseous molecules were simplified into single scattering processes. As a result, the MFASIS-simulated reflectance

was larger than the DOM-simulated reflectance, especially in dense cloud areas where the scattering interactions between clouds and gaseous molecules were non-negligible (Scheck et al., 2016).

## 5.4 Comparison with a previous study on FY-4A observing systems




The O-B biases were also explored for the FY-4A/AGRI channel 2 based on the CMA-MESO forecasts and RTTOV-DOM forward operator in a previous study by Zhou et al., (2024). In Zhou et al. (2024), the " $D_{\rm eff}$ " liquid cloud optical parameterization and the "Baran 2014" ice cloud optical parameterization were used. It is obvious that the configuration of cloud optical parameterization in Zhou et al. (2024) was sub-optimal. Nevertheless, the results in this study revealed many common characteristics with Zhou et al. (2024). For example, B consistently underestimated O on the domain average. In addition, an abrupt change was reported from 8 to 9 September 2020 by Figure 7 in Zhou et al. (2024), which was caused by the update of the calibration coefficients. In this study, an abrupt change of the O-B biases was also revealed from 31 March to 1 April for the four experiments (Figure 3a, Figure 10a, Figure 11a and Figure 12a). The abrupt changes in this study were more likely related to the abrupt change of the BRDF from March to April (Figure 13). In RTTOV, the BRDF of underlying land surface was taken from a monthly mean atlas, which is quasi-static and could not reflect the true temporal variation characteristics.

Figure 13. (a) Spatial distribution of the mean BRDF in March 2024; (b) Spatial distribution of the mean BRDF in April 2024.

One topic of Zhou et al. (2024) was to promote a bias-correction method based on the first-order approximation of the O-400 B bias. The findings in this study provide guidance for estimating more accurate B and the subsequent bias correction coefficient. Therefore, this study can be regarded as an extension of Zhou et al. (2024).

## 5.5 Influences of pre-assumed CCN number concentration





The pre-assumed CCN number concentration in equation (1) impacts  $D_{eff}$  for liquid water cloud. An increase in CCN number concentration distributes available water vapour among a greater number of aerosol particles, resulting in the formation of smaller cloud hydrometeors. Conversely, a reduction in CCN leads to fewer but larger hydrometeors, and vice versa. The typical CCN number concentration varies from 50, 100, 200, to 500 cm<sup>-3</sup> (Thompson et al., 2004). Accordingly, we conducted two additional experiments by configuring the CCN number concentration in Equation (1) as 100 cm<sup>-3</sup> and 500 cm<sup>-3</sup>. The sensitivity of TOA reflectance to the CCN number concentration is shown by Figure 14. For thin to moderately thick clouds (CWP < 3.2 (i.e., 10<sup>0.5</sup>) kg m<sup>-2</sup>, Figure 14), higher CCN concentrations yield reduced TOA reflectance, as smaller cloud hydrometeors enhance backscattering. In contrast, for optically thick clouds (CWP > 3.2 kg m<sup>-2</sup>), increased CCN number concentration results in greater TOA reflectance, likely due to more efficient multiple scattering processes. Nevertheless, the pre-assumed CCN number concentration demonstrates minimal impact on the evaluation results. The simulation results obtained with a CCN number concentration of 500 cm<sup>-3</sup> (Figure 15) show remarkable consistency with those generated using 300 cm<sup>-3</sup> (Figure 3). The results suggest that CCN's influence becomes less pronounced in synthetic visible satellite images. This occurs primarily because liquid water clouds are often overlain by upper-level ice clouds. Since visible-wavelength radiation has limited penetration depth through cloud layers, the observed reflectance is predominantly sensitive to the properties of the upper ice cloud layer rather than the underlying liquid water clouds affected by CCN. Consistent results were observed for the 100 cm<sup>-3</sup> CCN case (omitted for brevity), further demonstrating the robustness of our findings to variations in the prescribed CCN concentration.

Figure 14. Sensitivity of the reflectance, denoted by R, to the pre-assumed CCN number concentration, denoted by N. (a) Variation of reflectance with cloud water path (CWP) for different N. (b) Variation of reflectance difference with CWP for different N. The simulation was conducted for FY-4B band 2 by RTTOV-DOM, with the satellite zenith angle, satellite azimuth angle, solar zenith angle, and solar azimuth angle set to  $36\,^{\circ}$ ,  $0\,^{\circ}$ ,  $50\,^{\circ}$ ,  $55\,^{\circ}$ , respectively.

Figure 15. Time series of the O-B biases and standard deviations for (a-b) all, (c-d) cloudy, and (e-f) clear pixels. The settings of the experiment are similar to the CM-FY-DM experiment (CCN number concentration=300 cm<sup>-3</sup>), except that the CCN number concentration was set to 500 cm<sup>-3</sup> to calculate the effective diameter for liquid water cloud.

## 5.6 Influences of aerosol impact derived from the FY-4B aerosol products

In Equations (8) and (9), the aerosol impact was crudely identified by  $r_{aer}^{75}$  and  $r_{aer}^{25}$  for cloudy and cloud-free scenarios, respectively. In this part, the aerosol impact was further identified based on the FY-4B aerosol product. We conducted an extra experiment to account for aerosol impact on the evaluation results. The experiment settings were similar to the CM-FY-DM experiment, except that the aerosol optical properties were included in the RTTOV inputs when synthesizing visible satellite images. For a certain atmosphere column, the aerosol-related inputs to RTTOV include the extinction coefficient profile, the scattering coefficient profile, and the phase function at specified scattering angles. The aerosol extinction coefficient profile was derived from FY-4B aerosol optical depth (AOD) products at 650 nm (data quality flag = 3), assuming an exponential decrease in aerosol concentration with a scale height of 3 km. The assumption was based on the aerosol climatology over

China (Zhou et al., 2017), which revealed that the vertical distribution of aerosols commonly conforms to an exponential function and the typical value of scale height is around 3 km. In addition, sand dust is the most predominant aerosol type. Therefore, we used the phase function of sand dust aerosol type. The per-layer scattering coefficient was calculated by multiplying the extinction coefficient by the single scattering albedo of sand dust aerosol. The optical properties of the dust aerosol at the central wavelength of FY-4B AGRI band 2 were calculated by a logarithmic interpolation of the optical properties at 0.532 and 1.064 µm provided by Zhou et al. (2017).



445 Figure 16. Time series of the O-B biases and standard deviations for (a-b) all, (c-d) cloudy, and (e-f) clear pixels. The settings of the experiment are similar to the CM-FY-DM experiment (aerosols are ignored in RTTOV simulations), except that the aerosol impact was included in the RTTOV simulations to synthesize visible satellite images.

The FY-4B AOD products only provide valid retrievals under clear-sky conditions. Due to the cloud displacement errors of the CMA-MESO forecasts, the FY-4B AOD product provides aerosol optical properties over a pixel which is either cloudy  $(r_{sim} > r_{clr})$  or cloud-free for the CMA-MESO forecasts. Therefore,  $r_{aer}^{75}$  in Equation (8) was replaced by the mean differences of reflectance simulated with and without aerosol impact over cloudy pixels. Similarly,  $r_{aer}^{25}$  in Equation (9) was replaced by

the mean differences of reflectance simulated with and without aerosol impact over cloud-free pixels. Based on the experiment design, we re-calculated the biases and standard deviations of the O-B departure. The results indicate general consistency in cloudy conditions with the results for the CM-FY-DM experiment (Figure 3 and Figure 16). This aligns with established understanding, as aerosol influences become negligible under cloudy conditions, particularly with optically thick clouds. However, O was generally smaller than B in cloud-free conditions (Figure 16(e)), which is contrary to the results shown by Figure 3(e). Including aerosols in the RTTOV inputs for cloud-free conditions tend to increase the reflectance. Although the systematic underestimation of O relative to B in Figure 16(e) suggests a potential overestimation of aerosol impact in cloud-free conditions, the optimized cloud optical parameterizations derived from Figure 16 remain consistent with those shown in Figures 3, 10–12, and 15.

#### 6. Conclusions

The performance of RTTOV, a commonly used forward operator software package, is critical to the DA of satellite visible reflectance data in many aspects. During the radiative transfer simulations of RTTOV, cloud optical properties were determined by several built-in parameterizations in terms of cloud water content, cloud effective diameter, and ambient temperature. The radiative transfer is influenced by the cloud optical parameterizations. However, it is unclear which combination of liquid water cloud and ice cloud optical parameterizations reproduce the observed reflectance best.

In view of this problem, the performance of RTTOV under different cloud optical parameterizations was evaluated based on the observed and synthetic visible satellite images. To generalize the main findings, four experiments were performed. The experiments covered two observing systems including the FY-4B and Hiwawari-9 visible bands, two radiative solvers including DOM and MFASIS, and two atmospheric fields including the CMA-MESO forecasts and ERA5 data. Statistical characteristics of the O-B departure varied with the observing systems, the representativeness of the atmospheric fields, and the accuracy of radiative transfer modeling (e.g., scattering interactions between clouds and gaseous molecules). Nevertheless, consistent findings were revealed for different experiment designs.

In general, the O-B biases were sensitive to the cloud optical parameterizations. An analysis of one-month O-B biases revealed that the simulated reflectance was lower than observed reflectance. The smallest O-B bias was revealed for the liquid water cloud optical parameterization in terms of effective diameter (the " $D_{eff}$ " parameterization). This was in combination with the ice cloud optical parameterization developed by SSEC (the "SSEC" parameterization). For the "SSEC" parameterization, the effective diameter for ice clouds was parameterized in terms of ambient temperature and IWC, i.e., the B02 parameterization. Compared with the worst-performing cloud optical parameterization, the optimized configuration achieved a bias reduction of 0.02-0.04 on average, with only a marginal increase (

Baran, A. J., R. Cotton, K. Furtado, S. Havemann, L.-C. Labonnote, F. Marenco, A. Smith, and Thelen, J.-C.: A self-consistent scattering model for cirrus. II: The high and low frequencies, Q. J. R. Meteorol. Soc., 140, 1039-1057, doi: 10.1002/qj.2193, 2014.

Baum, B. A., Yang, P., Heymsfield, A. J., Schmitt, C., Xie, Y., Bansemer, A., Hu, Y. X., and Zhang, Z.: Improvements to shortwave bulk scattering and absorption models for the remote sensing of ice clouds, J. Appl. Meteorol. Clim., 50, 1037–1056, doi: 10.1175/2010JAMC2608.1, 2011.

- Boudala, F. S., Isaac, G. A., Fu, Q., and Cober, S. G.: Parameterization of effective ice particle size for high latitude clouds, Int. J. Climatol., 22, 1267-1284, doi: 10.1002/joc.774, 2002.
- Geiss, S., Scheck, L., de Lozar, A., and Weissmann, M.: Understanding the model representation of clouds based on visible and infrared satellite observations, Atmos. Chem. Phys., 21, 12273–12290, doi: 10.5194/acp-21-12273-2021, 2021.
  - Harnisch, F., Weissmann, M., and Periáñez, Á.: Error model for the assimilation of cloud-affected infrared satellite observations in an ensemble data assimilation system, O. J. R. Meteorol. Soc., 142: 1797–1808, doi:10.1002/qi.2776, 2016.
  - Hasselmann, K., Barnett, T. P., Bouws, E., Carlson, H., Cartwright, D. E., Enke, K., Ewing, J. A., Gienapp, H., Hasselmann, D. E., Kruseman, P., Meerburg, A., Müller, P., Olbers, D. J., Richter, K., Sell, W., and Walden, H.: Measurements of wind-
- wave growth and swell during the Joint North Sea Wave Project (JONSWAP), Dtsch. Hydrogr. Z., 8, 1–95, http://resolver.tudelft.nl/uuid:f204e188-13b9-49d8-a6dc-4fb7c20562fc, 1973 (last access: 23 September 2022).
  - Hersbach, H., Bell, B., Berrisford, P., Biavati, G., Horányi, A., Muñoz Sabater, J., Nicolas, J., Peubey, C., Radu, R., Rozum, I., Schepers, D., Simmons, A., Soci, C., Dee, D., and Thápaut, J.-N.: ERA5 hourly data on pressure levels from 1959 to present, Copernicus Climate Change Service (C3S) Climate Data Store (CDS) [data set], doi: 10.24381/cds.bd0915c6, 2018a.
- Hersbach, H., Bell, B., Berrisford, P., Biavati, G., Horányi, A., Muñoz Sabater, J., Nicolas, J., Peubey, C., Radu, R., Rozum, I., Schepers, D., Simmons, A., Soci, C., Dee, D., and Thápaut, J.-N.: ERA5 hourly data on single levels from 1959 to present, Copernicus Climate Change Service (C3S) Climate Data Store (CDS) [data set], doi: 10.24381/cds.adbb2d47, 2018b.
  - Hersbach, H., Bell, B., Berrisford, P., Hirahara, S., Horányi, A., Muñoz-Sabater, J., Nicolas, J., Peubey, C., Radu, R., Schepers, D., Simmons, A., Soci, C., Abdalla, S., Abellan, X., Balsamo, G., Bechtold, P., Biavati, G., Bidlot, J., Bonavita, M., De Chiara,
- G., Dahlgren, P., Dee, D., Diamantakis, M., Dragani, R., Flemming, J., Forbes, R., Fuentes, M., Geer, A., Haimberger, L., Healy, S., J. Hogan, R., Hólm, E., Janisková, M., Keeley, S., Laloyaux, P., Lopez, P., Lupu, C., Radnoti, G., de Rosnay, P., Rozum, I., Vamborg, F., Villaume, S., Thépaut, J.-N.: The ERA5 global reanalysis, Q. J. Roy. Meteor. Soc., 146, 1999–2049, doi: 10.1002/qi.3803, 2020.
- Hocking, J., Rayer, P., Rundle, D., Saunders, R., Matricardi, M., Geer, A., Brunel, P., and Vidot, J.:. RTTOV v12 Users Guide, 560 EUMETSAT Satellite Application Facility on Numerical Weather Prediction, p. 68, https://nwp-saf.eumetsat.int/site/download/documentation/rtm/docs\_rttov12/users\_guide\_rttov12\_v1.3.pdf, 2019 (accessed 27 December 2024).
  - Ji, W., Hao, X., Shao, D., Yang, Q., Wang, J., Li, H., and Huang, G.: A new index for snow/ice/ice-snow discrimination based on BRDF characteristic observation data, J. Geophys. Res. Atmos., 127, e2021JD035742, doi: 10.1029/2021JD035742, 2022.
- Kugler, L., Anderson, J. L., and Weissmann, M.: Potential impact of all-sky assimilation of visible and infrared satellite observations compared with radar reflectivity for convective-scale numerical weather prediction, Q. J. R. Meteorol. Soc., 149, 3623-3644, doi: 10.1002/qj.4577, 2023.
  - Lopez, P., Matricardi, M. and Fielding, M.: Validation of IFS+RTTOV/MFASIS solar reflectances against GOES-16 ABI observations. ECMWF Rechnical memorandum 893, doi:10. 21957/mprjictvg, 2022.

- Lopez, P., Matricardi, M.: Validation of IFS+RTTOV/MFASIS 0.64-μm reflectances against observations from GOES-16, GOES-17, MSG-4 and Himawari-8, ECMWF Technical memorandum 903, doi: 10.21957/l4u0f56lm, 2022.
   McFarquar, G., Iacobellis, M. S., and Somerville, R. C. J.: SCM simulations of tropical ice clouds using observationally based parameterizations of microphysics, J. Climate, 16, 1643-1664, doi: 10.1175/1520-0442(2003)016<1643:SSOTIC>2.0.CO;2, 2003.
- 575 Noh, Y.-C., Choi, Y., Song, H.-J., Raeder, K., Kim, J.-H., and Kwon, Y.: Assimilation of the AMSU-A radiances using the CESM (v2.1.0) and the DART (v9.11.13)–RTTOV (v12.3), Geosci. Model Dev., 16, 5365–5382, doi: 10.5194/gmd-16-5365-2023, 2023.
  - Okuyama, A., Takahashi, M., Date, K., Hosaka, K., Murata, H., and Tabata, T.: Validation of Himawari-8/AHI Radiometric Calibration Based on Two Years of In-Orbit Data, J. Meteorol. Soc. JPN, 96B, 91-109, doi: 10.2151/jmsj.2018-033, 2018.
- 580 Ou, S., and Liou, K.-N.: Ice microphysics and climatic temperature feedback, Atmos. Res., 35, 127-138, doi: 10.1016/0169-8095(94)00014-5, 1995.
  - Saunders, R., Hocking, J., Turner, E., Rayer, P., Rundle, D., Brunel, P., Vidot, J., Roquet, P., Matricardi, M., Geer, A., Bormann, N. and Lupu, C.: An update on the RTTOV fast radiative transfer model (currently at version 12), Geosci. Model Dev., 11(7), 2717-2737, doi: 10.5194/gmd-11-2717-2018, 2018.
- Saunders. R., Hocking, J., Turner, E., Havemann, S., Geer, A., Lupu, C., Vidot, J., Chambon, P., Köpken-Watts, C., Scheck, L., Still, O., Stumpf, C., and Borbas, E.: RTTOV-13 Science and validation report. https://nwp-saf.eumetsat.int/site/download/documentation/rtm/docs\_rttov13/rttov13\_svr.pdf, 2020 (last access: 27 December 2024). Scheck, L., Hocking, J., and Saunders, R.: A comparison of MFASIS and RTTOV-DOM, https://nwp-saf.eumetsat.int/publications/vs reports/nwpsaf-mo-vs-054.pdf, 2016 (last access: 23 December 2024)
- Scheck, L., Weissmann, M., and Bernhard, M.: Efficient Methods to Account for Cloud-Top Inclination and Cloud Overlap in Synthetic Visible Satellite Images, J. Atmos. Ocean. Tech., 35, 665-685, doi: 10.1175/JTECH-D-17-0057.1, 2018.
   Scheck, L., Weissmann, M., and Bach, L.: Assimilating visible satellite images for convective-scale numerical weather prediction: A case-study. Q. J. R. Meteorol. Soc., 146, 3165-3186, doi: 10.1002/qj.3840, 2020.
- Schröttle, J., Weissmann, M., Scheck, L., and Hutt, A.: Assimilating Visible and Infrared Radiances in Idealized Simulations of Deep Convection, Mon. Wea. Rev., 148, 4357-4375, doi: 10.1175/MWR-D-20-0002.1, 2020
  - Shen, X., Wang, J., Li, Z., Chen, D., and Gong, J.: China's independent and innovative development of numerical weather prediction, Acta Meteorologica Sinica (in Chinese), 78, 451-476, doi: 10.11676/qxxb2020.030, 2020.
  - Thompson, G., Rasmussen, R. M., and Manning, K.: Explicit forecasts of winter precipitation using an improved bulk microphysics scheme. Part I: Description and sensitivity analysis, Mon. Wea. Rev., 132(2), 519–542. doi: 10.1175/1520-0493(2004)132%3C0519:EFOWPU%3E2.0.CO;2, 2004.

600

Várnai, T., and Marshak, A.: Statistical Analysis of the Uncertainties in Cloud Optical Depth Retrievals Caused by Three-Dimensional Radiative Effects, J. Atmos. Sci., 58, 1540-1548, doi: 10.1175/1520-0469(2001)058<1540:SAOTUI>2.0.CO;2, 2001.

- Vidot, J., Baran, A. J., and Brunel, P.: A new ice cloud parameterization for infrared radiative transfer simulation of cloudy radiances: Evaluation and optimization with IIR observations and ice cloud profile retrieval products, J. Geophys. Res. Atmos., 120, 6937–6951, doi: 10.1002/2015JD023462, 2015.
  - Vidot, J., and Borb &, É.: Land surface VIS/NIR BRDF atlas for RTTOV-11: model and validation against SEVIRI land SAF albedo product, Q. J. R. Meteorol. Soc., 140, 2186-2196, doi: 10.1002/qj.2288, 2014.
  - Vidot, J., Brunel, P., Dumont, M., Carmagnola, C., and Hocking, J.: The VIS/NIR Land and Snow BRDF Atlas for RTTOV:
- 610 Comparison between MODIS MCD43C1 C5 and C6, Remote Sens., 10, 21, doi:10.3390/rs10010021, 2018.
  - Wan, Z., Wang, J., Huang, L., and Kang, J.: An improvement of the shallow convective parameterization scheme in the GRAPES-Meso, Acta Meterologica Sinica (in Chinese), 73, 1066-1079, doi: 10.11676/qxxb2015.07, 2015.
  - Wyser, K.: The effective radius in ice clouds, J. Climate, 11, 1793-1802, doi: 10.1175/1520-0442(1998)011<1793:TERIIC>2.0.CO;2, 1998.
- Yang, J., Zhang, Z., Wei, C., Lu, F., and Guo, Q.: Introducing the New Generation of Chinese Geostationary Weather Satellites, Fengyun-4, B. Am. Meteorol. Soc., 98, 1737-1658, doi: 10.1175/BAMS-D-16-0065.1, 2017.
  - Yao, B., Liu, C., Yin, Y., Zhang, P., Min, M., and Han, W.: Radiance-based evaluation of WRF cloud properties over East Asia: Direct comparison with FY-2E observations, J. Geophys. Res. Atmos., 123, 4613–4629, doi: 10.1029/2017JD027600, 2018.
- Zhang, X., Tang, W., Zheng, Y., Sheng, J., and Zhu, W.: Comprehensive Evaluations of GRAPES\_3 km Numerical Model in Forecasting Convective Storms Using Various Verification Methods, Meteorological Monthly (in Chinese), 46, 367-380, doi: 10.7519/j.issn.1000-0526.2020.03.008, 2020.
  - Zhang. B., Hu, X., Zhou W., Sha, J., and Chen, L.: Research on the radiometric calibration method for deep convective clouds in the reflective bands of FY-4A/AGRI, National Remote Sensing Bulletin (in Chinese), 1-11. doi: 10.11834/jrs.20243528,
- 625 2025
  - Zhou, Y.-B., Cao, T.-R., and Zhu, L.-J.: The processed data for evaluating cloud optical parameterizations in RTTOV, Zenodo [data set], doi: 10.5281/zenodo.14642334, 2025.
  - Zhou, Y.-B., Liu, Y.-B., Huo, Z.-Y., and Li, Y.: A preliminary evaluation of WRF (ARW v4.1.1)/DART (Manhattan release v9.8.0)-RTTOV (v12.3) in assimilating satellite visible radiance data for a cyclone case, Geosci. Model Dev., 15, 7397-7420,
- 630 doi: 10.5194/gmd-15-7397-2022, 2022.
  - Zhou, Y.-B., Liu, Y.-B., & Han, W.: Demonstrating the potential impacts of assimilating FY-4A visible radiances on forecasts of cloud and precipitation with a localized particle filter. Mon. Wea. Rev., 151, 1167-1188, doi: 10.1175/MWR-D-22-0133.1, 2023.
  - Zhou, Y.-B., Liu, Y.-B., Han, W., Zeng, Y-F., Sun, H.-F., Yu, P.-L., and Zhu, L.-J.: Exploring the characteristics of Fengyun-
- 4A Advanced Geostationary Radiation Imager (AGRI) visible reflectance using the China Meteorological Administration Mesoscale (CMA-MESO) forecasts and its implications for data assimilation, Atmos. Meas. Tech., 17, 6659-6675, doi: 10.5194/amt-17-6659-2024, 2024.

- Zhou, Y.-B., Sun, X.-J., Zhang, C.-L., Zhang, R.-W., Li, Y., and Li, H.-R.: 3D aerosol climatology over East Asia derived from CALIOP observations, Atmos. Environ., 152, 503–518, https://doi.org/10.1016/j.atmosenv.2017.01.013, 2017.
- Zhu, L., Gong, J., Huang, L., Chen, D., Jiang, Y., and Deng, L.: Three-dimensional cloud initial field created and applied to GRAPES numerical weather prediction nowcasting, J. Appl. Meteor. Sci., 28(1): 38-51, doi: 10.11898/1001-7313.20170104., 2017.