# Peer review of "Optimizing cloud optical parameterizations in Radiative Transfer for TOVS (RTTOV v12.3) for data assimilation of satellite visible reflectance data: an assessment using observed and synthetic images"

_EGUsphere, 2025_

## Author Comment (AC1)

**Response to Reviewer #3**

**Overall evaluations:**

The research is highly relevant as it focuses on optimizing cloud optical parameterizations in RTTOV for satellite visible reflectance data assimilation. With the increasing importance of satellite - based data in weather forecasting and climate studies, improving the accuracy of radiative transfer models like RTTOV is crucial. The use of data from recent geostationary meteorological satellites (FY - 4B and Himawari - 9) makes the study timely. The authors conducted four different experiments, covering two observing systems, two radiative solvers, and two atmospheric fields. A wide range of data was used, including forecasts from the CMA - MESO model, ERA5 reanalysis data, and observations from FY - 4B and Himawari - 9 satellites. The data was carefully processed, and multiple statistical methods were employed, such as analyzing O - B biases, standard deviations, and probability density functions. These analyses provided in - depth insights into the performance of different cloud optical parameterizations. The study identified the optimal cloud optical parameterization suite ("Deff" + "SSEC" + B02), which can significantly reduce the O - B biases. This finding has practical implications for improving the accuracy of radiative transfer simulations and data assimilation in weather prediction models.

**Our response:**

We would like to extend our sincere gratitude for your time and effort in reviewing our manuscript. We have carefully considered all your comments and suggestions and have made significant revisions to the manuscript accordingly. A point-by-point response addressing each of your concerns is attached below.

**Major issue 1:**

The number concentration in Eq. (1) was set to 300 cm$^{-3}$. Is this generally applicable for all types of clouds? If not, what is the sensitivity of the simulated reflectivity to the other number concentration?

**Our response:**

The number concentration of cloud droplets was set to 300 cm$^{-3}$ mainly because we want to maintain consistency with the commonly used microphysical schemes. On one hand, the default physic scheme of the CMA-MESO model for operational forecasting is the single-moment six-class (WSM6) microphysical scheme. The default number concentration of cloud droplets for the WSM6 scheme was set to 300 cm$^{-3}$. On the other hand, we have conducted the data assimilation of FY-4A visible reflectance under a framework of Observing System Simulation Experiments (OSSEs) (Zhou et al., 2022; 2023). In the OSSEs, the microphysical

scheme was set to the Thompson scheme. The default number concentration of cloud droplets for the Thompson scheme was also 300 cm$^{-3}$.

The number concentration of cloud droplets are determined by the pre-assumed Cloud Condensation Nuclei (CCN). An increase in CCN number concentration distributes available water vapour among a greater number of aerosol particles, resulting in the formation of smaller cloud hydrometeors. Conversely, a reduction in CCN leads to fewer but larger hydrometeors, and vice versa. According to Thompson et al. (2024), the typical CCN number concentration varies from 50, 100, 200, to 500 cm$^{-3}$. Therefore, we conducted two additional experiments to explore the sensitivity of the simulated reflectivity to the CCN number concentration. The two experiments were performed by configuring the CCN number concentration in Equation (1) as 100 cm$^{-3}$ and 500 cm$^{-3}$. The results are consistent with these with the CCN number concentration of 300 cm$^{-3}$. This is because although the reflectance shows some sensitivity to the pre-assumed CCN number concentration, the impact of the CCN number concentration is less pronounced in synthetizing visible images due to the radiative effects of the upper-layer ice clouds. (Line: 97-104; 402-428)

**Major issue 2:**

For cloud ice, is the effective diameter in Eq. (2) independent of cloud ice content and number concentration? What is the number concentration assumed for cloud ice?

**Our response:**

This is an interesting question actually. We checked the paper by Ou and Liou (1994). The ice crystal size distribution is determined by Equation (R1),

$$\mathrm{n(L)} = AL^B \cdot IWC \qquad \text{(R1)}$$

where A and B are empirical coefficients determined from measured data, and IWC denotes the Ice Water Content with a unit of g m$^{-3}$.

Ou and Liou (1994) conclude that A, B, and IWC depend on temperature. Therefore, $\mathrm{n(L)}$ and the effective diameter (Deff) of ice clouds are also functions of temperature. By theory, the number concentration of cloud ice, denotes by N, can be calculated by Equation (R2),

$$\mathrm{N} = \int_{L_{min}}^{L_{max}} \mathrm{n(L)} dL \qquad \text{(2)}$$

Therefore, Deff and IWC (or number concentration) are interrelated, with their relationship being temperature-dependent.

The authors explicitly give the formula to calculate Deff, which is the Equation (2) in our manuscript. Unfortunately, they do not provide an explicit formulate to describe $\mathrm{n(L)}$. Therefore, we cannot calculate the number concentration of cloud ice.

**Major issue 3:**

In CMA-MESO, there are six types of hydrometeors. How is the scattering from rain, snow and graupel is treated in this study?

**Our response:**

The operational configuration of the microphysical scheme for the CMA-MESO model is the single-moment six-class microphysical scheme (Hong and Lim, 2006). The model configuration generates five types of cloud hydrometeors, including cloud droplet, raindrop, ice, snow, and graupel.

RTTOV treats liquid water cloud and ice cloud with different methods. For liquid water clouds, cloud droplets and raindrops impact the radiative transfer simulations by tuning the extinction and scattering coefficients. This is done by multiplying the per-concentration (1 g m$^{-3}$) extinction and scattering coefficients for selected liquid water cloud parameterization by the total liquid cloud concentration (the sum of cloud droplets and raindrops). For ice clouds, ice, snow, and graupel impact the radiative transfer simulation in a similar way to the liquid water clouds. To be specific, the extinction and scattering coefficients are the product of per-concentration (1 g m$^{-3}$) extinction and scattering coefficients for selected ice cloud parameterization and the total ice cloud concentration (the sum of ice, snow, and graupel). (Line: 79-83; 101-104; 126-128)

**Major issue 4:**

Since FY-4B AGRI has an aerosol AOD product, it is better to use AOD to quantify the contribution from aerosol? Eq. (8) and (9) seems to be too objective.

**Our response:**

It true that Equations (8) and (9) in the manuscript are too objective since no observed aerosol information was considered. Following your guidance, we conducted an extra experiment to account for aerosol impact on the evaluation results. The experiment settings were similar to the CM-FY-DM experiment, except that the aerosol optical properties were included in the RTTOV inputs when synthesizing visible satellite images.

For a certain atmosphere column, the aerosol-related inputs to RTTOV include the extinction coefficient profile, the scattering coefficient profile, and the phase function at specified scattering angles. The aerosol extinction coefficient profile was derived from FY-4B aerosol optical depth (AOD) products at 650 nm (data quality flag = 3), assuming an exponential decrease in aerosol concentration with a scale height of 3 km. The assumption was based on

the aerosol climatology over China (Zhou et al., 2017), which revealed that the vertical distribution of aerosols commonly conforms to an exponential function and the typical value of scale height is around 3 km. In addition, sand dust is the most predominant aerosol type. Therefore, we used the phase function of sand dust aerosol type. The per-layer scattering coefficient was calculated by multiplying the extinction coefficient by the single scattering albedo of sand dust aerosol. The optical properties of the dust aerosol at the central wavelength of FY-4B AGRI band 2 were calculated by a logarithmic interpolation of the optical properties at 0.532 and 1.064 μm provided by Zhou et al. (2017).

It is noted that the radiative transfer simulations are rather time-consuming when the aerosol impact is included. For the results shown in Section 5.6 of the revised manuscript, it took about 15 days to complete the RTTOV simulations on our Linux cluster. Therefore, the aerosol impact was only discussed for the CMA-MESO 3-h forecasts, the RTTOV-DOM solver, and for the FY-4B visible band. Considering that the results for CM-FY-DM, E5-FY-DM, CM-HW-DM, and CM-HW-MF experiments are similar, the optimized cloud optical parameterizations derived from the aerosol-related experiment remain consistent with those shown the original manuscript. (Line:243-247; 429-460)

**Major issue 5:**

Captions for Figure 7b do not make sense. What is Fig 7c? Also, it is strange to show O-B bias from one month data.

**Our response:**

We are sorry for made a mistake here. Figure 7(a) denotes the spatial distribution of the one-month biases of O-B departure for the CM-FY-DM experiment. Figure 7(b) denotes the spatial distribution of the one-month standard deviation of O-B departure. Figure 7(c) denotes the spatial distribution of the one-month correlation coefficient between O and B.

We add the spatial distribution of one-month O-B statistics mainly to give some explanations to the discrepancies between O and B, which should be helpful to potential readers to better understand the errors in B. From the results shown by Figure 7, we conclude that the accuracy of B is not only determined by the cloud optical parameterizations, but also by the deficiencies of the CMA-MESO model in complex terrain areas and the inaccuracy of surface albedo of the RTTOV BRDF atlas in snow-covered areas.

We are uncertain whether it would be better to delete this section or retain it, so we would like to follow your recommendation in making the final decision. (Line: 302-317)

**Major issue 5:**

The DOM solver simplified the scattering interactions between clouds and gaseous molecules into single - scattering processes in non - cloudy layers. This simplification may lead to inaccuracies in the simulated reflectance, especially in areas with complex cloud - gas interactions. Although the MFASIS solver has some improvements in this regard, the overall treatment of scattering in the study could be more refined.

**Our response:**

We fully agree that accounting for multiple scattering interactions between clouds and gaseous molecules would further improve the accuracy of RTTOV simulations. However, we must acknowledge that addressing this issue currently lies beyond the scope of our expertise.

While these simplifications may introduce systematic errors in the simulated reflectance, previous studies (e.g., Scheck et al., 2020) have demonstrated that the current RTTOV framework still provides valuable contributions to the data assimilation of satellite visible reflectance data. Should this limitation be resolved in future developments, we would be eager to re-examine its impact on our evaluation results.

---

## Author Comment (AC2)

**Response to Reviewer #1**

**General comment:**

Review of the manuscript titled "Optimizing cloud optical parameterizations in RTTOV for data assimilation of satellite visible reflectance data: an assessment using observed and synthetic images" by Yongbo Zhou, Tianrui Cao, and Lijian Zhu.

This paper evaluates parameterizations of visible reflectance simulated from CMA-MESO and ERA5 using RTTOV with observations from instruments onboard the satellites FY-4B and Himawari-9.

Results show that the observed reflectance is higher than modelled reflectance, on average, and that the choice of an optimal configuration yields the lowest bias while only slightly increasing unbiased error (standard deviation of O-B). In clear-sky pixels, the atmospheric contribution to visible reflectance is near zero. Thus, the choice of parameterization had no effect. The bias is induced by the land surface representation in the NWP model. In cloudy sky, bias is more relevant than in clear sky cases, because the bias is responsible for a larger fraction of error, as shown in Figures 10, 11, and 12. In clear sky, bias was mostly lower than unbiased error, except of experiment CM-FY-DM in April (Figure 3).

The authors already nicely addressed my comments to an earlier version of the manuscript in review for another journal. For example, the authors extended the analysis by the standard deviation of O-B. The manuscript is now an extensive evaluation and in a very good condition. I recommend publication after minor comments are addressed.

**Our response:**

We would like to extend our sincere gratitude for your time and effort in reviewing our manuscript again. We carefully considered all your comments and suggestions and made revisions to the manuscript accordingly. We believe your review of this manuscript has greatly enhanced the clarity and rigor of our manuscript. A point-by-point response addressing each of your concerns is attached below.

**Minor comment 1:**

The increased B for C213 would amplify the O-B differences, leading to the increased standard deviations". The standard deviation (SD) is not influenced by the bias (average O-B). Thus, a change in average B does not change the SD. The explanation for an increased SD must be larger O-B differences, due to B being too high and too low (increase in |O-B|).

**Our response:**

Thank you for pointing this out. This part was re-written in the revised manuscript. (Line: 290-297)

The optical characteristics of the "Baran 2014 + B02" ice cloud parameterization may partially explain the observed phenomenon where the C213 parameterization successfully corrects the systematic underestimation of B while introducing greater standard deviation (or local variability) (Figure 3(a)-(d) in the revised manuscript). Within the same cloud water path (CWP) range, C213 produces the largest variation amplitude in B when transitioning from cloud-free to cloudy scenarios (Figure 5(b) in the revised manuscript). The impact of cloud optical parameterizations should be mainly determined by the upper ice clouds. The lower-layer liquid water clouds should be less pronounced in synthetizing visible images due to the radiative effects of the upper-layer ice clouds. Therefore, when transitioning from cloud-free to cloudy conditions, the "Baran 2014 + B02" scheme exhibits the most significant reflectance variation range (the yellow line Figure 5(b)). This enhanced variability in B could potentially broaden the distribution of O-B discrepancies, thereby increasing the standard deviation of O-B.

[Figure]

Figure 5. Dependence of FY-4B/AGRI band 2 reflectance on cloud water path (CWP) for different (a) liquid water cloud optical parameterizations and (b) ice cloud optical parameterizations. For this simulation, the solar zenith angle, viewing zenith angle, and relative azimuth angle are set to 25 °, 40 °, and 135 °, respectively.

**Minor comment 2:**

Figure 4: The caption seems to have an error, as the x-axis label is "reflectance" not "O-B departure".

**Our response:**

Thank you for pointing this out. We made a mistake here. It should be the "reflectance" rather than the "O-B" departure. (Line: 270)

**Minor comment 3:**

Equation 6 seems to be incorrect, not all terms are in the exponent, compared to Equation 2 of McFarquar et al. (2003).

**Our response:**

This is an interesting question. Actually this problem made me confused at the beginning of this work. The Equation in McFarquhar et al. (2003) is given by following equation,

$$r_e = 10^a + b\log(z) + c\log(z)^2$$

where $z = \text{IWC}/IWC_0 (IWC_0 = 1.0 \text{ g m}^{-3})$. a, b, and c are 1.78449, 0.281301, and 0.0177166 respectively. McFarquhar et al. (2003) also provided a figure to show the best fit (the solid line) of Deff as a function of IWC.

[Figure]

FIG. 2. Here $r_e$, defined from CEPEX in situ data as described in text, is presented as a function of IWC, also derived from in situ data. Each dot represents 10 s (or 2 km) averaged-size distribution. Solid line represents best fit to data, thin long dashes and thin short dashes represent plus and minus one and two std devs from mean relation (see text for details). Thick lines represent relationships determined by McFarquhar (2001): vertical bars, $-65°$; dashes, $-55°$; dashes and vertical bars, $-45°$; dashes and 3 vertical bars, $-35°C$.

However, in the RTTOV codes (${RTTOV}/src/main/rttov_profaux.F90), Deff was calculated by the following methods,

```
         ELSE IF (prof(iprof)%idg == 3) THEN
           ! Scheme by Boudala et al., 2002, Int. J. Climatol., 22, 1267-1284.
           ztempc           = prof(iprof)%t(lev) - rtt
           aux%ice_dg(lay, iprof) = 53.005_jprb * ((prof_int(iprof)%cloud(6, lay)) ** 0.06_jprb) * EXP(0.013_
jprb * ztempc)
         ELSE IF (prof(iprof)%idg == 4) THEN
           ! Scheme by McFarquhar et al. (2003)
           amcfarq          = 1.78449_jprb
           bmcfarq          = 0.281301_jprb
           cmcfarq          = 0.0177166_jprb
           zmcfarq          = prof_int(iprof)%cloud(6, lay)
           aux%fac1_ice_dg(lay, iprof) =      &
             & 10.0_jprb ** (amcfarq + (bmcfarq * LOG10(zmcfarq)) + (cmcfarq * LOG10(zmcfarq) * LOG10(zmcfarq
)))
```

If we translate the codes in to a formula form, it is the Equation (6) in our manuscript. Therefore, there much be some errors in either Equation (6) or the Equation (2) in McFarquhar

et al. (2003). This problem can be easily solved by comparing the Deff-IWC function with the Fig. 2 in McFarquhar et al. (2003). We did such a simulation and the result is shown by Fig. R1. Therefore, we believe there should be a typo error of the Equation (2) in McFarquhar et al. (2003).

[Figure]

Figure R1. The Deff-IWC function derived from the Equation (6) in our manuscript and the Equation (2) in McFarquhar et al. (2003)

**Minor comment 4:**

L237, L275: Language: Instead of "was expected", you probably mean "was found"? Or was there an expectation before seeing the results?

**Our response:**

This is a misunderstanding due to our poor English skill. The word "expected" was replaced by "revealed" or "found". (Line: 256; 288)

**Minor comment 5:**

L243: You state "opposite circumstance" but the same sign (O-B>0) as in the line above. Probably you mean O-B < 0?

**Our response:**

Thank you for pointing this out. You are correct. Since we re-wrote this part. The sentences you mentioned were deleted in the revised manuscript. (Line: 290-297)

**Minor comment 6:**

L248: "at the low- or high-reflectance ends": Maybe you can quantify this, e.g. "at low reflectance (<0.1)" or similar.

**Our response:**

Corrected. (Line: 264)

**Minor comment 7:**

L271: "The results in Figure 4 suggested that the Baran 2018 ice scheme should be used with caution ...". I am confused, because Figure 4 shows results for ice_scheme=1 (SSEC) and not for "Baran" parametrization. Maybe you meant Figure 5?

**Our response:**

Yes you are correct. The information is given by Figures 3 and 5. The error was corrected in the revised manuscript. (Line: 284)

**Minor comment 8:**

Figure 7 caption (c) missing.

**Our response:**

Corrected. (Line: 315-317)

**Minor comment 9:**

L367: Language, "B was consistently underestimated compared with O", I think this should be "B consistently underestimated O".

**Our response:**

Corrected. (Line: 390)

**Minor comment 10:**

L393: "The smallest O-B bias". Good, but can you quantify it? For example "the optimal configuration reduced the bias by 0.04 on average, while standard deviation increased by less than 0.005.

**Our response:**

Corrected. (Line: 479-480)